# Unveiling Temporal Telltales: Are Unconditional Video Generation Models Implicitly Encoding Temporal Information?

## Abstract

Unconditional video generation models seemed to generate realistic videos. However, in this paper, we delve into what could be the meaning of 'realness' in the video generation models. Similar to human observers, we expected Convolution Neural Networks (CNNs) to struggle in classifying the temporal location of generated videos using a single frame due to the limited temporal information a single frame alone provides. However, our preliminary experiments unveil that current unconditional video generation models actually do inadvertently encode temporal location into each frame, enabling CNNs to correctly classify the temporal location of generated videos. To alleviate such a problem, we propose a method by adding the Gradient Reversal Layer (GRL) with lightweight CNN to the prior works to explicitly neglect this implicitly encoded temporal information. The experimental results, indeed, show that the implicit encoding of temporal information while training the unconditional video generator does negatively influence the FVD score. Moreover, experiments on diverse prior video generation models and datasets show that our method can be used in a plug-and-play manner. Also, the results show the successful elimination of implicitly encoded temporal information without compromising the FVD score, highlighting the need to consider temporal classification accuracy as a supplementary metric in video generation models.

## 1 Introduction

Image and video generation models have recently attracted significant attention, primarily due to their remarkable success in generating realistic samples. However, how could we validate the "realness" of generated samples? Numerous research in image generation tasks employ metrics such as Fréchet Inception Distance (FID) (Heusel et al., 2017), Learned Perceptual Image Patch Similarity (LPIPS) (Zhang et al., 2018), Mean Absolute Error (MAE), Structural Similarity Index Measure (SSIM), and more to quantitatively measure how closely the generated samples align with real-world samples. Likewise, the quantitative evaluation of video generation tasks utilizes the Fréchet Video Distance (FVD) (Unterthiner et al., 2018) to measure the fidelity of the generated samples. As these evaluation metrics are meticulously designed, they offer a precise quantitative measurement as generated samples with higher or lower values (depending on the metric) align well with human perceived quality, especially with FID and FVD (Skorokhodov et al., 2022). In addition to the quantitative results, generative research utilizes qualitative results. More importantly, researchers frequently leverage user studies (Tulyakov et al., 2018; Shen et al., 2023; Lezama et al., 2022; Kim et al., 2022; Kwon et al., 2022) to provide additional justification for the realisticity of generated samples. The additional qualitative results and user study exhibit factors beyond numerical metrics to assess what **appears natural to human observers**, a crucial aspect considering the intended alignment of artificial intelligence with human perception.

One often overlooked characteristic of video is that each independent frame from the unseen generated videos cannot be temporally classified by humans. In essence, a single frame from a video does not provide sufficient information on its temporal location within the video. Consequently, when human observers are tasked with classifying temporal locations of frames from random videos, their accuracy in classifying the correct temporal location is comparable to random guessing. For exam-

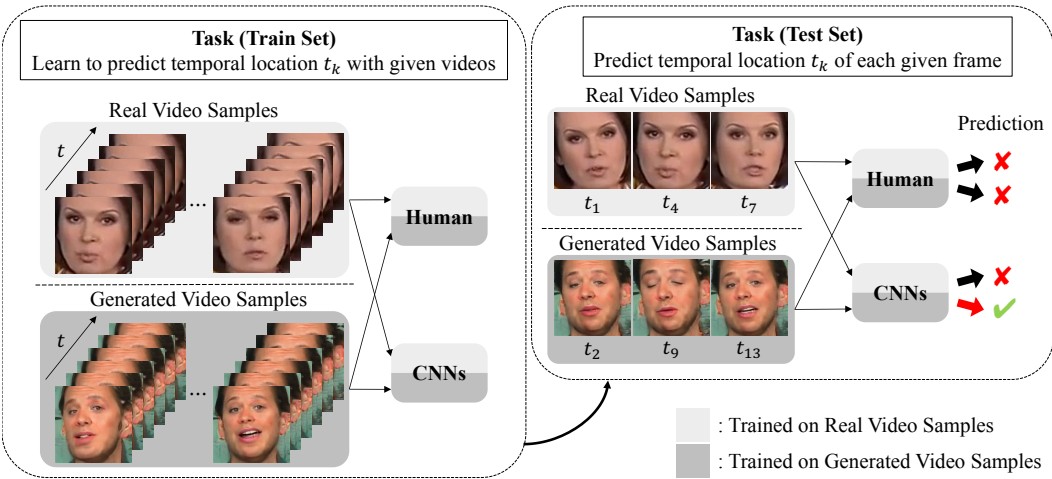

Figure 1: Convolutional Neural Networks (CNNs) can classify the temporal location of frames in videos generated by unconditional video generation methods. This behavior is even more intriguing and can be viewed as a problem, given that the model struggles to accurately classify the temporal location when presented with real-world videos. For example, we have four sets of videos: train/test sets of real-world videos and videos generated by the unconditional video generator. We first present the train sets of real-world videos to humans and neural networks to learn the temporal location of videos. Then, if we ask them to localize the temporal location of randomly chosen frames from the test sets of real-world videos, they are unable to temporally classify them correctly. However, when the same process is applied to the generated videos, CNNs demonstrate remarkable temporal localization accuracy while humans struggle, as observed in parallel with the results from the real-world videos.

ple, in the case of generated videos consisting 16 frames, human observers' accuracy of getting the right temporal location would result in around $16/100 = 6.25\%$. Then how about CNNs? These networks exhibit similar behavior with the real-world videos, unable to classify the temporal location correctly. However, it's intriguing that CNNs exhibit different behaviors when classifying generated videos. In generated videos, they often demonstrate an impressive ability to precisely classify and sometimes perfectly identify the temporal location as depicted in Figure 1. These unintended capabilities of the temporal classification of the neural networks come from unconditional video generation models implicitly encoding the temporal information during training and sampling. The additional temporal information misguides the distribution of the generated video to deviate away from the real-world video distribution. Then would CNNs generate more realistic videos in terms of feature distance if this implicitly encoded temporal information is alleviated?

In this paper, we first present an experiment with real-world videos and generated videos with lightweight CNN , ResNet-18, to highlight the model's intriguing behavior in classifying the temporal location of the generated videos. Through this experiment, we argue for the first time that current unconditional video generation models inadvertently encode temporal information into each frame when training the generator, enabling correct classification of temporal location with CNNs. This highlights the need to consider the subtle characteristics (i.e., temporal information not present in the frames) of real-world videos when training the video generation model. Furthermore, we demonstrate through experiments that the encoded temporal information negatively influences the FVD score. Thus, we explicitly neglect encoded temporal information in each frame with a simple method using the Gradient Reversal Layer (GRL) added to each unconditional video generation model with a ResNet-18 in a plug-and-play manner. Experiment results demonstrate that our method decreases temporal classification accuracy, implying successful neglect of the temporal information while showing better or comparable FVD performance. Our contribution can be summarized as follows:

- We demonstrate that current unconditional video generation models are generating videos without considering the characteristics of real-world video through experiments.

- To disregard the implicitly encoded temporal information within each frame, we propose a simple method using GRL with lightweight CNN and experimentally show that temporal information has been erased from the video through temporal classification accuracy.

- We argue that unconditional video generation models should consider temporal classification accuracy as a supplementary metric.

- Experiment results with our method demonstrate that neglecting implicitly encoded temporal information does not adversely affect the generated video quality, as indicated by better or comparable FVD score.

## 2 PRELIMINARY - PROBLEM STATEMENT

In this section, we outline our preliminary experiments on CNNs' performance in localizing temporal frames within real-world and generated video samples. We anticipate CNNs to perform similarly to random guessing. Through this experiment, we confirm that CNNs do struggle in classifying temporal locations with real-world videos like humans but are able to easily classify with generated video samples.

The experiments are conducted with three video generation benchmarks: FaceForensics (Rössler et al., 2018), Sky–Timelapse (Xiong et al., 2018), and UCF-101 (Soomro et al., 2012). First, we reconstruct each dataset for training/testing the temporal classifier. For each dataset, we randomly select a single content category (e.g., v_BasketballDunk_g01_c01 in the case of UCF-101). This is a more viable and understandable situation, as frames would be grouped by their respective categories in the feature space when multiple categories are utilized rather than being grouped by their temporal location. Within the chosen category, we fix the $0^{th}$ frame as the first frame for all 2,048 videos. Then we randomly sampled an additional 15 samples from the video to make a 16 frame video. This process of selecting 16 frames within the same category was repeated 2,048 times to construct a dataset. Finally, 2,048 videos are split into 80%/20% for training/testing. We utilized ResNet-18, a widely recognized and proven architecture, for our temporal classifier. The only modification done on the ResNet-18 is the number of outputs of the fully connected layer which was 16, the number of frames. The experiments for the real-world videos and generated videos shared the same hyperparameter setting of 150 epochs, 0.001 learning rate, and 64 batch size. The generated videos utilized in this experiment were produced using a reproduced MoCoGAN (Tulyakov et al., 2018). In contrast to the original MoCoGAN setting, which generates $64 \times 64$ frames, we trained MoCoGAN with $256 \times 256$ images. Consequently, the generated videos were also $256 \times 256$ in size, ensuring a fair comparison with the real-world videos, which are $256 \times 256$.

In Table 1, we present the train and test accuracy in real-world videos and generated videos. In conventional classification tasks, high accuracy is desired, but for temporal classification, we aim for low accuracy to mimic human prediction capabilities. Surprisingly, CNNs exhibited the ability to correctly classify temporal locations of generated videos, contrary to our expectations given their foundation in human visual neuroscience. Notably, CNNs achieved significantly higher test accuracy when classifying generated videos compared to real-world videos. In addition to the test accuracy, we observed the accuracy in the training phase. The training accuracies of generated videos are substantially higher than those of real-world videos. This suggests the existence of temporal features in the generated samples which are only detectable by CNNs. Finally, one could argue that tuning hyperparameters when training real-world videos may result in higher temporal accuracy. We have tried multiple hyperparameter settings for the real-world videos but the train and test accuracy re-

Table 1: The temporal accuracy result of three benchmarks. The training and testing accuracies with real-world videos highlight the difficulty in accurately classifying temporal locations when randomly chosen frames are given.

| Dataset | Temporal (%) | | | | | |
| | **FaceForensics** | | **SkyTimelapse** | | **UCF-101** | |
| | Train | Test | Train | Test | Train | Test |
| Real-World Videos | 24.61 | 4.19 | 26.62 | 9.47 | 29.05 | 15.72 |
| Generated Videos | 99.45 | 87.70 | 99.71 | 78.80 | 99.45 | 81.68 |

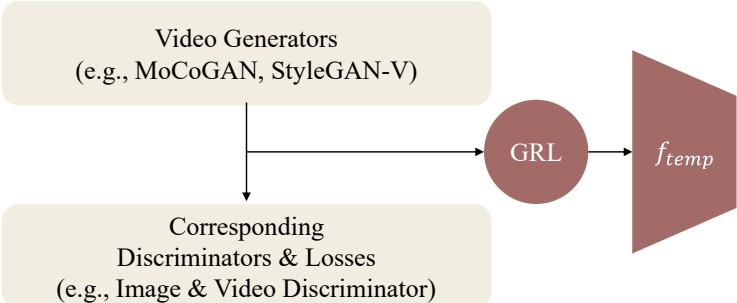

Figure 2: The overall flow of our method to explicitly decode the temporal information when training. Within each unconditional video generation method, we integrate a Gradient Reversal Layer (GRL) along with an ImageNet pre-trained ResNet-18 model, denoted as $f_{temp}$. All the losses employed in each method are maintained for all of our experiments with our method.

mained around the values presented in Table 1. This highlights the absence of temporal features in the case of real-world videos.

## 3 METHOD

### 3.1 OVERVIEW

As previously highlighted, videos generated by unconditional video generation models possess a distinct characteristic: they implicitly encode temporal information that can be recognized by CNNs. To eliminate such a characteristic, we propose a straightforward baseline method to explicitly decode the temporal information while training the unconditional video generator.

Generally, unconditional video generation methods (Tulyakov et al., 2018; Wang et al., 2020; Tian et al., 2021; Skorokhodov et al., 2022; Yu et al., 2022) are given content latent vector $z_c$ and motion latent vectors $[z_m^{(0)}, ..., z_m^{(t)}]$ as the input for the generator $G_{orig}$, and generate videos by

$$V = G_{orig}(\{z_c, z_m^{(0)}\}, ..., \{z_c, z_m^{(t)}\}). \tag{1}$$

The $G_{orig}$ is conventionally adversarially trained by the discriminator $D_{orig}$ and additional losses. However, the architecture and components can vary across methods. Some methods incorporate a pre-trained image generator, while other might modify the two discriminators (image and video discriminators). Therefore, the loss function may vary as depicted in Figure 2.

In our method, we adopt an adversarial training technique using GRL with a simple network. More specifically, we train a temporal classifier $f_{temp} : I \rightarrow [0, 1]^T$ with a ResNet-18 architecture to estimate a temporal location from each image in conjunction with the discriminator, where $I$ represents each frame of generated videos and $T = \{0, ..., t\}$ represents the temporal length of the videos. When training the generator, GRL adds a negative cross-entropy loss between the time positions estimated by $f_{temp}$ and the actual time positions to the existing generator training loss to prevent the generator from encoding temporal information.

### 3.2 PREVENT IMPLICIT ENCODING OF TEMPORAL INFORMATION

To mitigate the inadvertent encoding of temporal information within unconditional video generation methods, we propose a method consisting of GRL with the temporal classifier $f_{temp}$ which classifies each frame to the right temporal location. The GRL is commonly used in domain adaptation (Ganin et al., 2016; Huang et al., 2020; Choi et al., 2022) to output a similar effect with adversarial training but with the classifier. GRL achieves this by leaving the input unchanged during the forward propagation but reversing the gradient by multiplying it by a negative scalar during the backpropagation. Therefore, for GRL to work properly trainable inputs must be present for the direction of the gradient to be opposite from the correct labels. In the case of current video generation methods, temporal

classification of frames from generated videos is easily performed, which allows GRL to reverse the gradient and for the generator to produce videos that are not temporally classifiable.

For temporal classifier $f_{temp}$, an ImageNet pre-trained ResNet-18 is employed. The generated video $V_i = \{I_i^{(0)}, ..., I_i^{(t)}\}$ are labeled according to temporal class and utilized to train the $f_{temp}$, where $i$ represents the $i$-th generated video. The training of the temporal classifier takes place simultaneously with the training of the discriminator, and it is trained to distinguish the temporal location of frames within videos generated by the generator. The loss function for $f_{temp}$ becomes:

$$\mathcal{L}_{temp} = \frac{1}{n \cdot (t+1)} \sum_{i \in N, j \in T} \mathcal{L}_{CE}(f_{temp}(I_i^{(j)}), j), \tag{2}$$

where $N = \{1, ..., n\}$ represents the number of generated videos and $T = \{0, ..., t\}$ represents the temporal length of generated videos. Note that we generated $n$ number of videos with fixed content noise vector for $f_{temp}$ to be able to extract and learn the temporal information within the generated videos.

To alleviate the implicitly encoded temporal information from the generator, negative cross-entropy loss from GRL with the temporal classifier is added to the original generator loss to help the generator to be temporally confused. The generator loss function and other additional loss functions utilized in each method are denoted as $\mathcal{L}_{orig}$ as we did not modify them. The loss function for the generators becomes:

$$\mathcal{L}_{gen} = \mathcal{L}_{orig} - \lambda \cdot \mathcal{L}_{temp}. \tag{3}$$

The $\lambda$ represents a scalar and the negative is placed due to the GRL. Through adversarial training between the discriminator, temporal classifier, and generator, it is possible to train an unconditional video generation model that prevents the encoding of temporal information during video generation. The proposed method can be simply added to existing video generation methods in a plug-and-play manner. The full framework of the proposed method is shown in Figure 2.

## 4 EXPERIMENTS

### 4.1 EXPERIMENT SETTINGS

#### 4.1.1 DATASET

- **FaceForensics (Rössler et al., 2018)** consists of 704 news videos with various reporters. We have cropped and extracted each frame following the procedure outlined in the StyleGAN-V (Skorokhodov et al., 2022). The final resolution of the video is $256 \times 256$.

- **Sky-Timelapse (Xiong et al., 2018)** consists of dynamic sky scenes such as sunset or moving clouds. For all of our experiments, we utilized 2,114 videos, each with a resolution of $256 \times 256X$. These videos were preprocessed according to the StyleGAN-V (Skorokhodov et al., 2022)

- **UCF101 (Soomro et al., 2012)** is commonly used for video action recognition task. It includes 13,220 videos of 101 different action categories of size $320 \times 240$. Each frame is cropped to $240 \times 240$ and resized to $256 \times 256$. In all of our experiments, both train and test videos are employed.

#### 4.1.2 METRICS

Following prior works, we report Fréchet Video Distance (FVD) (Unterthiner et al., 2018) and temporal accuracy, first introduced in this paper. We utilize the FVD implementation provided by the StyleGAN-V (Skorokhodov et al., 2022) which reduced the discrepancies in the evaluation protocols used in the previous works. FVD is measured with 2,048 videos of 16 frames each from the real and generated videos. Temporal accuracy measures the precision of locating random frames within the generated videos in terms of their temporal position. For the temporal accuracy metric, we utilize the same checkpoint used for computing the FVD to generate another set of 2,048 videos of 16 frames. Note that videos generated for temporal accuracy are distinct from those used for computing FVD as we generated them by fixing content latent vectors but varying motion latent

Table 2: Quantitative results on FaceForenscis dataset. Lower values for temporal accuracy and FVD score indicate better performance as denoted by the ↓. **Bold** indicates the better performance between the original work and our method (+ Ours).

| Method | FaceForensics | |
|---|---|---|
| | Temporal (%) (↓) | $FVD_{16}$ (↓) |
| MoCoGAN $64^2$ | 30.51 | 407.76 |
| + Ours | **12.63** | **369.35** |
| MoCoGAN $256^2$ | 87.70 | 1658.79 |
| + Ours | **5.27** | **1522.24** |
| MoCoGAN-HD $256^2$ | 90.63 | 178.52 |
| + Ours | **7.91** | **177.12** |
| StyleGAN-V $256^2$ | 11.36 | 103.38 |
| + Ours | **8.25** | **90.90** |

Table 3: Quantitative results on Sky-Timelapse dataset. Lower values for temporal accuracy and FVD score indicate better performance as denoted by the ↓. **Bold** indicates the better performance between the original work and our method (+ Ours).

| Method | Sky-Timelapse | |
|---|---|---|
| | Temporal (%) (↓) | $FVD_{16}$ (↓) |
| MoCoGAN $64^2$ | 28.15 | 434.28 |
| + Ours | **15.61** | **409.69** |
| MoCoGAN $256^2$ | 80.84 | 1551.87 |
| + Ours | **70.18** | **1007.77** |
| MoCoGAN-HD $256^2$ | 99.99 | 529.76 |
| + Ours | **97.25** | **477.09** |
| StyleGAN-V $256^2$ | 8.26 | **86.31** |
| + Ours | **5.61** | 91.92 |

vectors. We have split the 2,048 videos into 80%/20% for train/test and utilized to train a pre-trained ResNet-18, not the temporal classifier $f_{temp}$ utilized for training the generator. All the experiments for temporal accuracy share the same hyperparameter setting of 150 epochs, 0.001 learning rate, and 64 batch size. The only hyperparameter for our method is $\lambda$ which is scheduled by the current training epochs divided by the total epochs for each video generation model.

### 4.1.3 BASELINE & IMPLEMENTATION DETAILS

We report three baselines, MoCoGAN (Tulyakov et al., 2018), MoCoGAN-HD (Tian et al., 2021), and StyleGAN-V (Skorokhodov et al., 2022), to demonstrate the effectiveness of eliminating temporal information embedded in the frames. The three baselines are selected for their groundbreaking contribution to the field of unconditional video generation. More specifically, MoCoGAN disentangled the motion and content vector and successfully generated videos. MoCoGAN-HD enabled the generated videos to be in a higher resolution and StyleGAN-V treated the motion vector as the continuous signal and also modified the structure of the discriminator.

For all experiments, we used the open-source implementation provided by each author and we re-trained each method to clearly demonstrate the effectiveness of our method. The MoCoGAN-HD and StyleGAN-V are trained using $256 \times 256$ videos to generate videos of the same resolution. In the case of MoCoGAN, we trained two models by varying the resolution of training videos, $64 \times 64$ and $256 \times 256$, as the original work was conducted using $64 \times 64$. This is done because MoCoGAN is not designed to generate high-resolution videos and to better demonstrate that temporal information is implicitly encoded regardless of the resolution. We trained MoCoGAN on a single RTX2080Ti GPU and trained MoCoGAN-HD and Stylegan-V with 4 A6000 GPUs. The hyperparameter settings for both the baseline and the GRL attached models have not been further tuned to achieve better results.

Table 4: Quantitative results on UCF-101 dataset. Lower values for temporal accuracy and FVD score indicate better performance as denoted by the ↓. **Bold** indicates the better performance between the original work and our method (+ Ours).

| Method | UCF-101 | |
| --- | --- | --- |
| | Temporal (%) ($\downarrow$) | FVD$_{16}$ ($\downarrow$) |
| MoCoGAN $64^2$ | 32.91 | 2539.05 |
| + Ours | **6.43** | **2360.25** |
| MoCoGAN $256^2$ | 82.36 | 4890.48 |
| + Ours | **11.42** | **4589.31** |
| MoCoGAN-HD $256^2$ | 96.72 | 1729.43 |
| + Ours | **28.22** | **1466.90** |
| StyleGAN-V $256^2$ | 10.20 | 1684.94 |
| + Ours | **7.19** | **1546.40** |

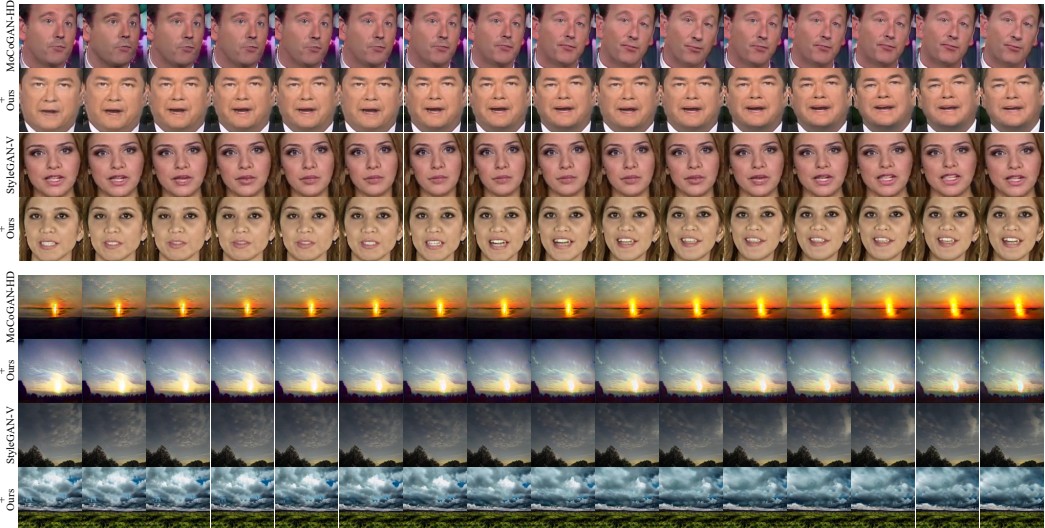

Figure 3: Random generated video samples from the two existing methods, MoCoGAN-HD and StyleGAN-V, and each with our method on FaceForensics and Sky-Timelapse dataset. These are the generated 16-frame video clips. Zoom in for the best view.

## 4.2 RESULTS

### 4.2.1 QUANTITATIVE RESULTS

In Tables 2,3, and 4, we present the performance of temporal classification accuracy and FVD. The $FVD_{16}$ denotes the FVD score calculated with 16 frames. $64^2$ and $256^2$ represent the resolution of the videos utilized for training and generated video samples. Our method, employing $f_{temp}$ and GRL, is denoted as + Ours. In all of the experiments with diverse unconditional video methods and dataset, we consistently achieved lower temporal accuracy while obtaining improved FVD scores except for one experiment. This demonstrates the effectiveness of our method which can be integrated into various unconditional video generation methods in a plug-and-play manner.

In Table 3, StyleGAN-V achieves a better FVD score than StyleGAN-V + Ours, however, it is negligible as it only differs by 5. It is notable that in many cases the temporal accuracy drop is substantial when original works are compared with the + Ours, especially in MoCoGAN-HD. This may be caused by the MoCoGAN-HD mapping the trajectory of the motion rather than sampling. As the motion is treated as the trajectory, the distance between the $0^{th}$ frame and others may be the cue for the temporal location. In the case of StyleGAN-V, we observed that the temporal accuracy is around random guessing. This could be due to StyleGAN-V utilizing the holistic discriminator

rather than two separate discriminators, image and video discriminators. As holistic discriminator is conditioned on the time distances between frames, they were able to suppress the implicit temporal encoding. However, as our method further reduces the effect of inadvertently encoded temporal information from each frame the temporal accuracy of StyleGAN-V + Ours decreases and makes the FVD score go down.

### 4.2.2 QUALITATIVE RESULTS

In Figure 3, we provide generated samples from the MoCoGAN-HD and StyleGAN-V, and each with our method on two datasets, FaceForensics and Sky-Timelapse. As our method is building upon each prior work, and because our method does not have any additional method that enhances the video quality, the generated samples within the same method (i.e., MoCoGAN-HD and MoCoGAN-HD + Ours) are not much different. However, as there are no vivid visual differences between each other, we conclude that applying temporal classifier $f_{temp}$ with the GRL layer does not harm the visual quality of the video. Thereby, we contend that we have only eliminated the temporal information embedded in each frame from the generated video samples through our quantitative and qualitative results.

## 5 RELATED WORKS

In the early stage of the unconditional video generation task, many research focused on disentangling the content and motion from the video. Likewise, VGAN (Peng et al., 2019) utilized GAN to generate a foreground scene using 3D deconvolution and combined it with a 2D background mask to create a video. Then, MoCoGAN (Tulyakov et al., 2018) and TGAN (Xu & Veeramachaneni, 2018) disentangled the motion and content vector with 2D image generator and RNN structured network. This was done by fixing the content vector while varying the motion vectors to produce content-consistent videos. Then, G3AN (Wang et al., 2020) proposed a 3D spatio-temporal generator network that gets content and motion noise vector at the same time to generate the video. MoCoGAN-HD utilized pre-trained StyleGAN (Karras et al., 2019) and proposed to predict a sequence of latent motion trajectory by training a motion generator for producing the content-consistent frames in the video. StyleGAN-V (Skorokhodov et al., 2022) proposes the motion vector as a continuous signal and utilizes modified discriminators for the generator to better understand the motion. DIGAN (Yu et al., 2022) building upon INR-GAN (Skorokhodov et al., 2021) treated not only the motion vector but also the content vector as the continuous signal. As StyleGAN-V and DIGAN treated the motion vector as the continuous signal, they were able to stably generate a longer video than the MoCoGAN-HD. Recently, there have been attempts to utilize diffusion models for video generation (Luo et al., 2023; Yu et al., 2023a; Harvey et al., 2022) and they show comparable performance with the Generative Adversarial Network (GAN) based approaches.

## 6 CONCLUSION AND DISCUSSION

In this paper, we highlight that current unconditional video generation methods are generating videos that implicitly encode the temporal information within the video. This unexpected characteristic allows the CNNs to classify the temporal location when presented with random frames from the generated videos, a task in which CNNs struggle with real-world videos. To alleviate such a problem, we propose a simple yet effective method utilizing the Gradient Reversal Layer (GRL) with the ImageNet pre-trained ResNet-18. Our method explicitly trains the generator to eliminate temporal information in an adversarial manner. However, there is a limitation. As we are utilizing the classification models, it cannot be trained for long videos (i.e., videos lasting 1 hour) as there would be too many frames to consider for classification. For future work, architectural advancements that do not necessitate classification during training may be considered.

The experiment results demonstrate the success of eliminating temporal information, significantly lowering the temporal classification accuracy of the generated videos while lowering the FVD score at the same time. Moreover, the diverse experiment setting shows the potential of using our method as a plug-and-play module with any unconditional video generation models. A notable result from the experiment is the temporal accuracies of StyleGAN-V which are already significantly low when compared with MoCoGAN and MoCoGAN-HD. This emphasizes the importance of considering

motion as a continuous signal and additionally modifying the discriminator to account for relationships between frames rather than solely relying on the image and video discriminators. We suggest utilizing temporal classification accuracy as a supplementary metric in the unconditional video generation field.

Finally, we acknowledge a concern shared with image generation models, the possibility of misusing video generation models for unethical purposes such as generating fake news videos. Our method inadvertently eliminates one potential cue for detecting fake videos and intensifying the threat. However, the quality of generated videos solely with the unconditional video models still exhibits noticeable disconnections when viewed as video. Nonetheless, the necessity of proactive research on detecting fake videos remains important.

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

# A  APPENDIX

## A.1  ADDITIONAL IMAGES

We present additional sample images of UCF-101 generated from MoCoGAN-HD, MoCoGAN-HD + Ours, StyleGAN-V, and StyleGAN-V + Ours in Figure 4. We also present the samples utilized for temporal classification in Figure 5.

## A.2  DIFFUSION-BASED UNCONDITIONAL VIDEO GENERATION MODEL

There are two streams in unconditional video generation models: GAN-based and diffusion-based. In this section, we demonstrate that implicit temporal encoding also exists in the diffusion-based model. For the experiment, we have experimented with a recent diffusion-based unconditional video generation model, PVDM (Yu et al., 2023b), in two benchmarks, Sky-Timelapse and UCF-101. We utilized the author-provided checkpoints for generating the videos in each dataset. Similar to the experiments with GAN-based models, we fix the content noise and vary the motion noise for generating 2,048 videos. As shown in Table 5, diffusion-based models also seem to implicitly encode the temporal information when generating the videos. Although the results of UCF-101 seem negligible, we argue that an improvement in the quality of the generated UCF-101 videos could correspond to an increase in temporal accuracy. The qualitative results of PVDM can be seen in Figure 6.

Table 5: The temporal accuracy result of two benchmarks, Sky-Timelpase and UCF-101.

| Dataset | Temporal (%) | |
| --- | --- | --- |
| | **Sky-Timelapse** | **UCF-101** |
| Real-World Videos | 9.47 | 15.72 |
| PVDM (Diffusion-based Generated Videos) | 45.55 | 19.66 |

## A.3  DEEPFAKE DETECTION

Deepfake videos have potential negative impacts in various aspects by disseminating false information worldwide through the Internet. In response to this threat, numerous deepfake detection algorithms have been recently researched. As the temporal classification metric showed a substantial difference between the real and generated videos, we have further experimented with the deepfake videos. We employed the Celeb-DF (Li et al., 2020) and Celeb-DF-v2 (Li et al., 2020) as they are commonly utilized in deepfake detection. Both datasets consist of real and synthesized videos while the synthesized datasets in each dataset are made by swapping the face from the real video with another face identity from another video with an auto-encoder. For a fair comparison between real and synthesized video, we first randomly select a video from real videos. Then, we select, the same video with another face identity in the synthesized video. In line with our original experiments, we fix the first frame as the $0^{th}$ frame from each video and randomly sample the rest 15 frames from the video. We note that the 15 frames are of the same index, meaning that they are from the same temporal location.

The temporal classification accuracy of the synthesized video is around 1.07%p higher than real video results. As the difference is not as notable as the experiments in the main tables, the temporal classification metric cannot be directly used for detecting deepfake videos. We have also experimented with the face-cropped version of each video, considering that synthesized videos are generated solely by face swapping the original video. We obtained the face-cropped version utilizing the Haar Cascade algorithm and sample images are illustrated in Figure 7. Temporal classification accuracy with the face-cropped version also showed a similar trend of 0.92%p higher with the face-cropped synthesized videos. We hypothesize that the small temporal accuracy difference comes from not generating entire videos but rather only detecting and swapping faces, thereby leaving the temporal characteristics of the real video unchanged.

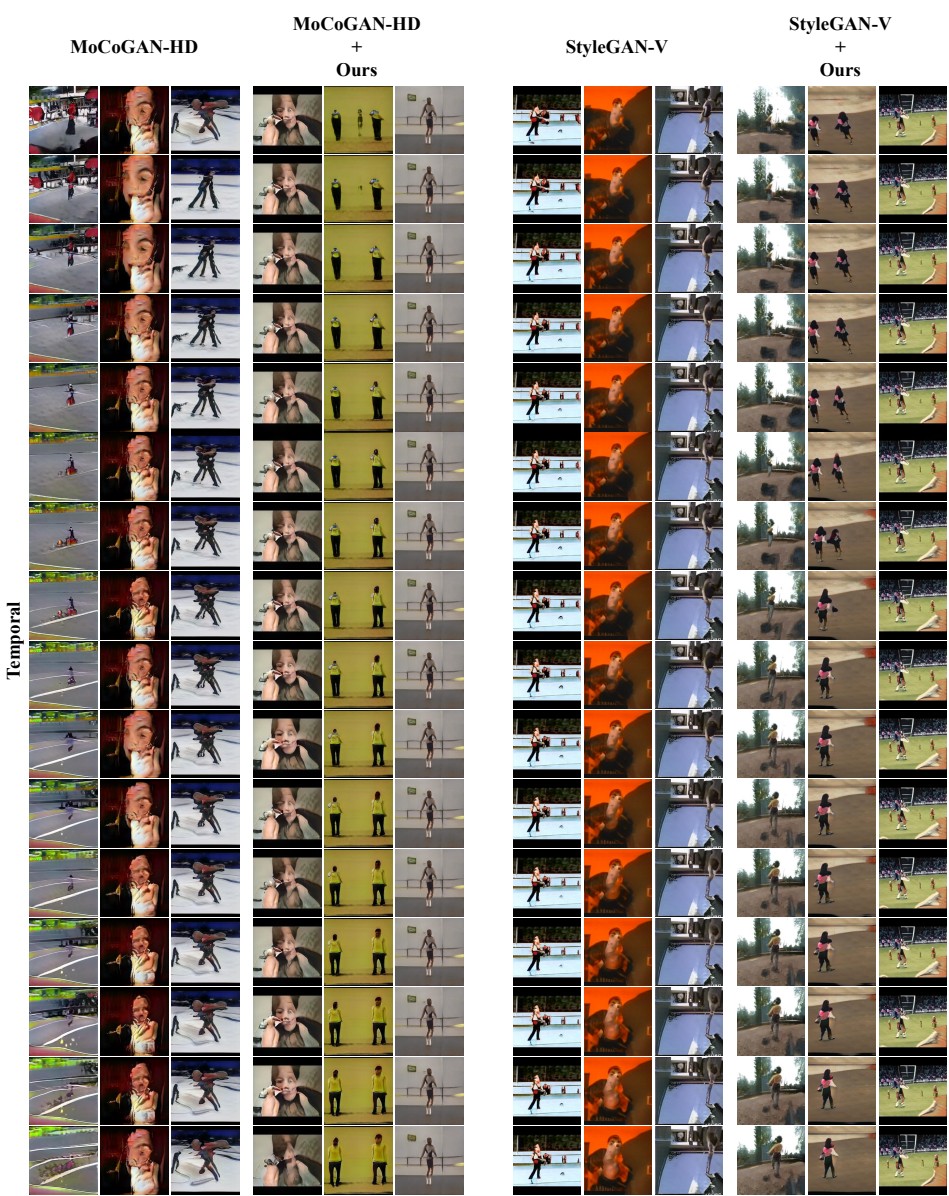

Figure 4: Sample images of UCF-101.

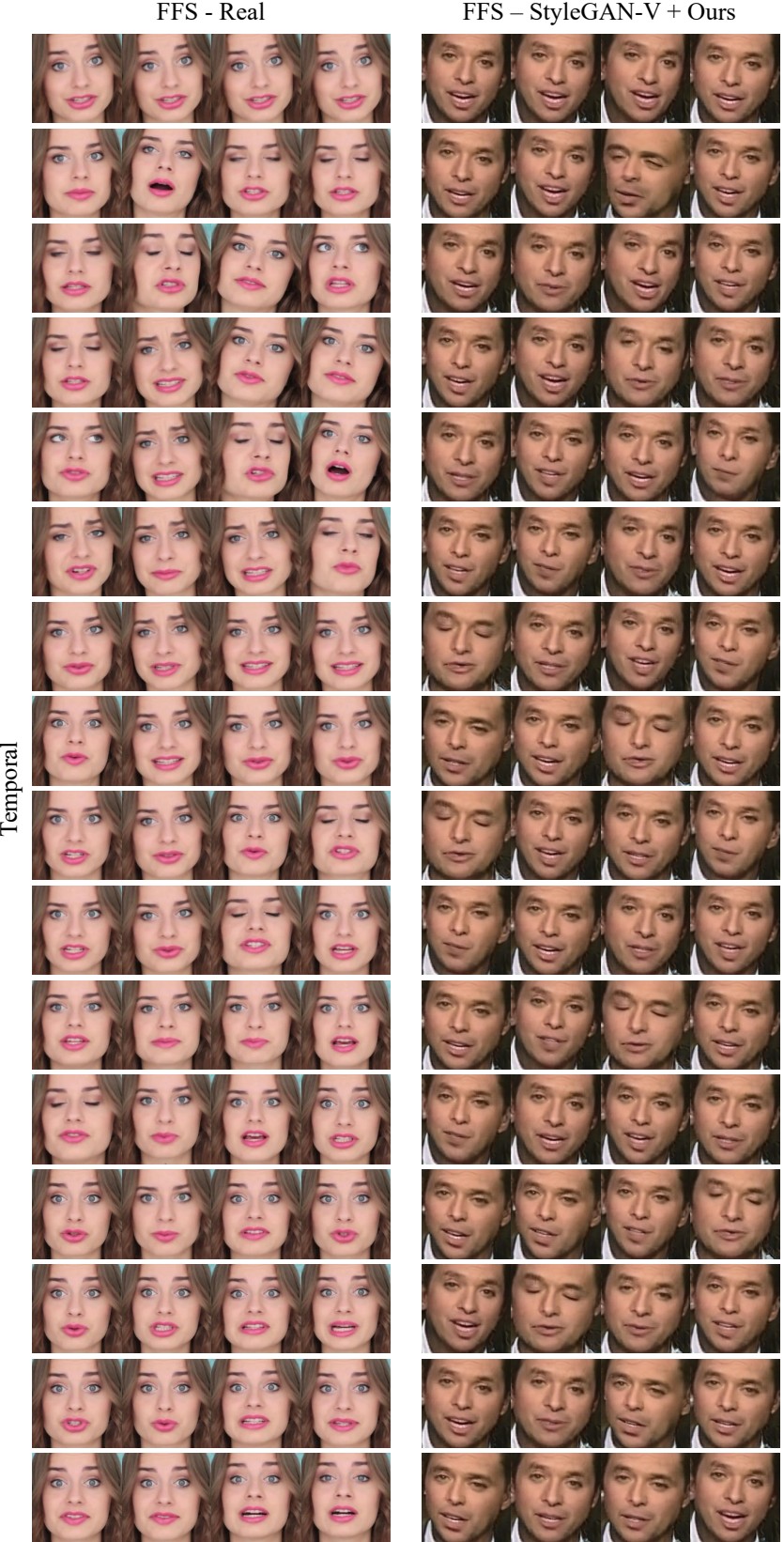

Figure 5: Sample images of temporal classification.

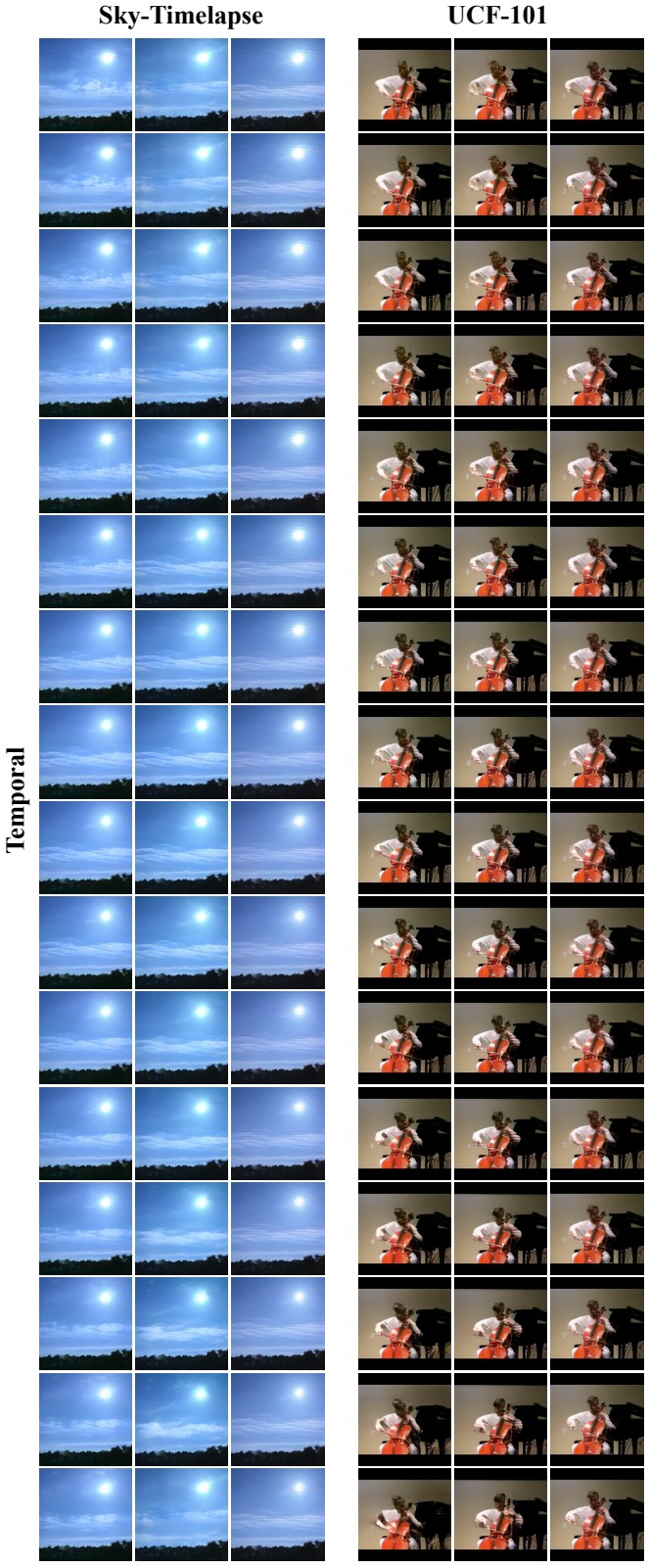

Figure 6: Generated videos of PVDM (Yu et al., 2023b) by fixing the content but varying the motion.

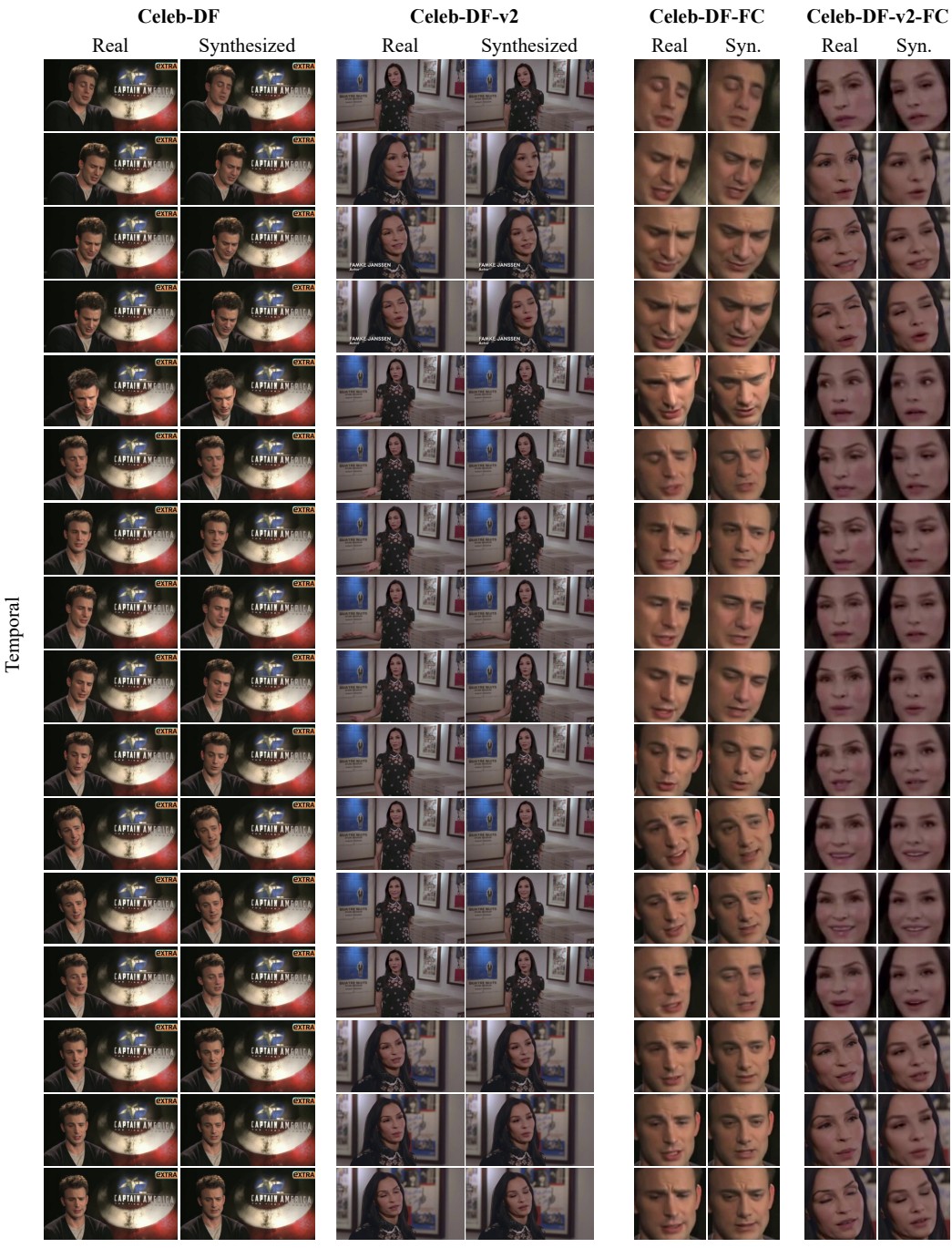

Figure 7: Sample images of Celeb-DF (Li et al., 2020), Celeb-DF-v2 (Li et al., 2020), and each corresponding face-cropped version.

