# OpenReview forum: "Unveiling Temporal Telltales: Are Unconditional Video Generation Models Implicitly Encoding Temporal Information?"
_ICLR.cc/2024/Conference — Submitted to ICLR 2024_

### Official Review · Reviewer_isck · 2023-10-23

**Soundness:** 2 fair
**Presentation:** 2 fair
**Contribution:** 2 fair
**Rating:** 5
**Confidence:** 4

**Summary:**

The authors propose a method to prevent unconditional video generation models from encoding temporal location in their outputs. The motivation for this modification is that humans do not rely on temporal location to classify dynamic visual information. The authors show that reducing the bias to encode temporal location often improves generation performance on video quality metrics aligned with human perception. Through their work, they argue that the ability to classify temporal location based on output videos could be used as an evaluation metric for video generation.

**Strengths:**

The authors make an interesting observation that temporal location is implicitly encoded in the outputs of unconditional video generation models. The experiment where they show that CNNs struggle to classify temporal information in real videos, but succeed on generated videos is an intriguing contribution and well motivated the preliminary problems statement.  Finally, the proposed method for reducing the bias to encode temporal location appears novel and sound to me.

**Weaknesses:**

I found the general argument of the work to be quite confusing. The logic came across as "humans cannot identify temporal location from one video frame, so it should not be possible to classify video frames from a good quality generative model." In general, it is difficult to justify how doing poorly at a task would improve a model. I think it would have been more effective and interesting to analyze *what other features* humans pay attention to when processing spatiotemporal information if not temporal location.

Although the authors emphasized that CNNs were inspired by human neuroscience, this was confusing as there was little discussion of how this work contributes to our understanding of CNNs as models of human visual processing. There was also little justification for why CNNs are good models of human spatiotemporal processing. Standard CNNs, in fact, have been shown to often be mis-aligned with human perception when looking at the temporal property perceptual straightness (Toosi \& Issa 2023, Harrington et al. 2023). The classification experiment, however, was still interesting, and I think it could be made stronger by de-emphasizing the human angle and focusing on the fact that temporal location is much easier to classify in generated videos than real.

Finally, the organization of the paper was a bit odd at times. I do not understand why the related work section came before the discussion. I also think the discussion could be expanded on, especially in thinking about what information humans use in video perception if not temporal location.

**Questions:**

As I touched on in the weakness, are the authors trying to make a statement on how human perception relates to CNNs? Or is it more about making classifying temporal location a human-inspired video quality metric? If it is more about the metric, did the authors consider analyzing a wider set of models or even running a human experiment to validate their results?

In general, I think the work could be strengthened by thinking what other features of human spatiotemporal perceptual could your work give us insight into other than the lack of temporal location encoding? Although I see notable weakness, I think there is a lot of potential in this work and would like to hear more from the authors about what they are hoping to convey about human perception and video generation through their work.

---

> ### Author Response · Authors · 2023-11-17
> **Author Response to Reviewer isck**
>
> Thank you again for having taken the time to thoroughly evaluate our submitted manuscript.
>
> In reaction to this, we went at great lengths to thoroughly address and accommodate every single one of your comments.
>
> We would very much appreciate you going through our responses to your comments, as well as the revised version. We hope that you are able to reconsider your evaluation and potentially provide additional feedback for a final version of the paper.
>
> Many thanks in advance for your time and efforts.
>
> ---
>
> Thank you for pointing out the part where readers could get confused. We have de-emphasized the human angle and focused more on the fact that temporal location is much easier to classify in the generated video than real in the revised version.
>
>
> In the result section we introduced a metric that should be considered when evaluating the generated videos since real-world videos are not temporally classifiable while generation tasks aim at generating realistic videos, not videos that can be temporally detected easily.
>
>
> Lastly, there is no clear answer for what features human spatiotemporal perceptually view. Generally speaking, when humans are presented with a video, we focus more on making a video that could contextually make sense. In other words, we look for the relationships between the frames. On the other hand, through our findings, temporal classifiers are not acting in the same way.

---

> > ### Comment · Reviewer_isck · 2023-11-23
> > **thank you for the response**
> >
> > I appreciate the revision you made to the paper. It reads a bit more clearly to me.
> >
> > Given your response, however, I still struggle to fully grasp the significance of your results. The performance improvements still seem a bit marginal, and there is no clear answer about what spatiotemporal features *should* be used for naturalistic generative videos.
> >
> > I keep my score a 5, but I welcome further comments.

---

> ### Author Response · Authors · 2023-11-23
> **Additional Response to Reviewer isck**
>
> It is great to hear that some of the initial concerns have been dismissed after our first response.
>
> Regarding the choice of features for generating naturalistic videos, we emphasize the significance of considering 'viewing time as a continuous signal.' In the context of this paper, the removal of temporal information proves crucial for generating realistic videos along the temporal axis. We assert that current unconditional video generation models primarily prioritize visual appeal without delving into the fundamental challenges of video generation.
>
> In this paper, we contend that the lack of realism in current unconditional video generation stems from the encoding of temporal information in each frame. As the author, I emphasize that the spatiotemporal feature crucial for generating naturalistic videos is the temporal information derived from real videos. This information can be extracted through a simple network or an RNN, but it must align with real-world statistics and characteristics. As mentioned in the original manuscript, we believe that treating temporal information/features as a continuous signal, as demonstrated in previous research [1], is instrumental in addressing implicit temporal encoding.
>
> [1] Skorokhodov, Ivan, Sergey Tulyakov, and Mohamed Elhoseiny. "Stylegan-v: A continuous video generator with the price, image quality and perks of stylegan2." _Proceedings of the IEEE/CVF Conference on Computer Vision and Pattern Recognition_. 2022.

---

### Official Review · Reviewer_kbs5 · 2023-10-30

**Soundness:** 2 fair
**Presentation:** 2 fair
**Contribution:** 2 fair
**Rating:** 3
**Confidence:** 4

**Summary:**

This paper proposes to integrate the Gradient Reversal Layer (GRL) into unconditional video generation, aiming at preventing the encoding of temporal location into each frame's features. This stems from the observation that humans struggle to classify the temporal location of a frame, while CNNs show impressive temporal classification accuracy on generated video frames. The experiments indicate that explicitly training unconditional video generation models to disregard the temporal information in the frames results in reduced temporal classification accuracy, while maintaining comparable or improved Frechet Video Distance (FVD) performance.

**Strengths:**

- The exploration of the relationship between temporal classification accuracy and the quality of unconditional video generation is novel and intriguing.
- Preliminary experiment results show the effort to unveil this novel insight, although the experimental design requires refinement.
- The proposed approach enhances several GAN-based unconditional video generation methods concerning the FVD metric.

**Weaknesses:**

1. The preliminary experiment design needs refinement:
- Experiments should encompass multiple baselines to bolster the claim that current unconsitional video generation methods implicitly encode temporal information.
- The constructed dataset used for training the temporal classifier appears overly homogenous since the clips are sampled from a single video clip of FaceForensics, Sky–Timelapse, or UCF-101.
- The labels for certain frames seem less meaningful due to the frame's presence in various clips at different positions, arising from the repeated random sampling during the construction of the temporal classification dataset.

2. Figure-related issues require attention:
- Figure 2 lacks an introduction of f_{temp} in the caption, and the caption references an ImageNet pre-trained model not presented in the figure.
- An invalid figure reference in the last paragraph of Section 3 suggests a missing figure in the manuscript.

3. The training quality of the reproduced MoCoGAN-256 appears suboptimal. Tables 2, 3, and 4 reveal an extremely high FVD value for MoCoGAN with 256x256 compared to the other high-resolution video generation baselines.

4. The validity of the negative gradient provided by the temporal classifier during training needs reconsideration. The temporal classifier is trained using frames from different videos, which is different from the practice in the preliminary experiment and the evaluation stage that constrains the training frames to have the same content.

5. The absence of recent diffusion models for video generation (Luo et al., 2023; Yu et al., 2023; Harvey et al., 2022) in the experiments diminishes the contribution of this work.

6. Additional insights on designing architectures that "do not necessitate classification" in the discussion section would be beneficial.

**Questions:**

- Why not use MoCoGAN-HD and StyleGAN-V in the preliminary experiment? This would avoid reproducing MoCoGAN-256.

- Can you visualize some real and generated video frames used in the preliminary experiment? This could enhance clarity.

- What is the reason for using distinct videos for evaluating temporal accuracy and FVD computation? Why not use the temporal classifier from the training process to evaluate temporal classification accuracy?

- Can you provide qualitative comparisons on UCF101? This would provide additional insights.

- Can you provide some failure cases of video generation with GRL?

---

> ### Author Response · Authors · 2023-11-17
> **Author Response to Reviewer kbs5 (1 / 3)**
>
> Thank you again for having taken the time to thoroughly evaluate our submitted manuscript.
>
> In reaction to this, we went at great lengths to thoroughly address and accommodate every single one of your comments.
>
> We would very much appreciate you going through our responses to your comments, as well as the revised version. We hope that you are able to reconsider your evaluation and potentially provide additional feedback for a final version of the paper.
>
> Many thanks in advance for your time and efforts.
>
> ---
>
> (Q1 & W1-1) Why not use MoCoGAN-HD and StyleGAN-V in the preliminary experiment? This would avoid reproducing MoCoGAN-256. / Experiments should encompass multiple baselines to bolster the claim that current unconditional video generation methods implicitly encode temporal information.
>
> (Response – Q1 & W1-1) The results in the preliminary section can be extended if we bring MoCoGAN-HD and StyleGAN-V results in the experiment section (Table. 2, 3, 4). However, we only presented the MoCoGAN results to avoid redundancy. Nevertheless, we agree that it would be better bolstered if we tested on diverse models (i.e., Diffusion-based model). As suggested by another reviewer, we have tested on recent diffusion-based unconditional video generation model to show the problem of implicit temporal encoding does not occur only in the GAN-based models but also in the diffusion-based model. Similar to the GAN-based model, the diffusion-based model also showed the existence of the problem by being able to temporally classify the generated videos. We have added the section in the revised appendix for this experiment (45.55% in Sky-Timelapse and 19.66% in the UCF-101).
>
>
> (W1-2&3 & Q2) The constructed dataset used for training the temporal classifier appears overly homogenous since the clips are sampled from a single video clip of FaceForensics, Sky–Timelapse, or UCF-101. / The labels for certain frames seem less meaningful due to the frame's presence in various clips at different positions, arising from the repeated random sampling during the construction of the temporal classification dataset. / Can you visualize some real and generated video frames used in the preliminary experiment? This could enhance clarity.
>
> (Response – W1-2&3 & Q2) As we fix the content noise vector while varying the motion noise vector, the generated videos consist of frames that look similar to another frame located in another temporal location and are also overly homogenous. With these generated videos, the CNNs are able to classify the temporal location of the frames better than the real-world videos. To mitigate the same dataset construction, we have meticulously designed the real-world dataset for temporal classification. We have added some samples for temporal classification in the revised appendix.
>
>
> (W2-1 & 2) Figure 2 lacks an introduction of f_{temp} in the caption, and the caption references an ImageNet pre-trained model not presented in the figure. / An invalid figure reference in the last paragraph of Section 3 suggests a missing figure in the manuscript.
>
> (Response – W2-1&2) Thank you for identifying places where additional explanations were needed and for pointing out typos. We have added the explanation in Figure 2 for the f_{temp} in the revised version. The ImageNet pre-trained model that we utilized is inside the torchvision package provided by PyTorch and there is no exact reference for this. However, we did add that we utilized ImageNet pre-trained ResNet-18 in the caption of Figure 2. The invalid figure reference in the last paragraph of Section 3 was supposed to refer to Figure 2 in the manuscript, we have corrected the reference in the revised version.
>
>
> (W3) The training quality of the reproduced MoCoGAN-256 appears suboptimal. Tables 2, 3, and 4 reveal an extremely high FVD value for MoCoGAN with 256x256 compared to the other high-resolution video generation baselines.
>
> (Response – W3) MoCoGAN was the first unconditional video generation model that disentangled the content and motion noise vector for generating videos. It is important to note that the primary focus of MoCoGAN was not centered on generating high-resolution videos such as 256x256 but on generating 64x64 videos. In the MoCoGAN paper, they experimented on the Weizmann Action dataset and UCF-101 resized to 64x64. Therefore, the MoCoGAN-256 could seem suboptimal when compared to MoCoGAN-HD and StyleGAN-V, the methods aimed at generating 256x256 videos.

---

> ### Author Response · Authors · 2023-11-17
> **Author Response to Reviewer kbs5 (2 / 3)**
>
> (W4) The validity of the negative gradient provided by the temporal classifier during training needs reconsideration. The temporal classifier is trained using frames from different videos, which is different from the practice in the preliminary experiment and the evaluation stage that constrains the training frames to have the same content.
>
> (Response – W4) A batch of generated videos for training the temporal classifier, different from the videos utilized for training the discriminator, is generated with fixed content noise as noted by the reviewer. We added an additional explanation in the revised version.
>
>
> (W5) The absence of recent diffusion models for video generation (Luo et al., 2023; Yu et al., 2023; Harvey et al., 2022) in the experiments diminishes the contribution of this work.
>
> (Response – W5) We admit that the problem has to be general across different backbones (GAN-based model and Diffusion-based model). From the suggested methods, we have tested PVDM[1] as they have an open-source implementation with the author-provided checkpoints. Due to the time constraint of the discussion period, we could not train the PVDM and PVDM + ours to demonstrate the effectiveness of our method. However, we did an experiment to see if the implicit temporal encoding is also occurring in the diffusion-based model. For the experiment, we generated the videos in Sky-Timelapse and UCF-101 with the checkpoint provided by the author. Consistent with our experiments in the manuscript, we have fixed the content noise vector while varying the motion noise vector, and maintain other experimental settings identical to the original experiment. The table below shows the temporal classification accuracy of the generated videos. The results demonstrate that implicit temporal encoding also occurs in the diffusion-based model. We have added the section in the appendix in the revised version with the visual samples of the generated videos that were used in the experiment.
>
> | Dataset | Sky-Timelapse | UCF-101 |
> | --- | --- | --- |
> | Real-World Videos | 9.47% | 15.72% |
> | Generated Videos (PVDM) | 45.55% | 19.66% |
>
>
> (W6) Additional insights on designing architectures that "do not necessitate classification" in the discussion section would be beneficial.
>
> (Response – W6) We sincerely appreciate the reviewer for raising this point. This is definitely a go-to way for future research. As stated in the discussion section, employing a classification model may pose challenges when generating extensively long videos (i.e., 1-hour or longer videos). We may try to make the classification model to learn the relative temporal location, however, this may be suboptimal or even not work properly. Unfortunately, we do not currently have any insights on architecture that do not necessitate classification. If possible, it would be a great honor to hear back the thoughts on the potential architecture construction.
>
>
> (Q3) What is the reason for using distinct videos for evaluating temporal accuracy and FVD computation? Why not use the temporal classifier from the training process to evaluate temporal classification accuracy?
>
> (Response – Q3) The FVD metric measures the feature distance similarity between the real-world videos and generated videos. Recognizing the suboptimal quality of early-stage generated videos, we employed a fully trained unconditional video generation model for generating images for measuring FVD and temporal accuracy.
> As for the temporal classifier, we did not use the one from the training as the reviewer stated. While the temporal classifier was designed to accurately classify the temporal location of frames during training, the inclusion of the Gradient Reversal Layer (GRL) before the temporal classifier may have harmed its classification ability. To ensure precise measurement of the temporal accuracy we re-trained the temporal classification model.
> To make a fully functional temporal classification, one would need a sufficient amount of data. In this work, we generated 2,048\times0.8 videos for training.
>
>
> (Q4) Can you provide qualitative comparisons on UCF101? This would provide additional insights.
>
> (Response – Q4) The quality of the generated UCF-101 videos hampers from comparing the results visually. Prior works also show the Sky-Timelapse and FaceForensics for qualitative comparison as done in our paper. However, we added some UCF-101 results in the revised appendix to provide additional insights to the reviewer.
>
>
>  [1] Yu, Sihyun, et al. "Video probabilistic diffusion models in projected latent space." Proceedings of the IEEE/CVF Conference on Computer Vision and Pattern Recognition. 2023.

---

> ### Author Response · Authors · 2023-11-17
> **Author Response to Reviewer kbs5 (3 / 3)**
>
> (Q5) Can you provide some failure cases of video generation with GRL?
>
> (Response – Q5) We acknowledge the importance of presenting failure cases for advancing research in this field. However, we have not observed any failure cases while training. However, a potential failure scenario arises when the temporal classifier struggles to classify temporal location during training. Specifically, if the generated videos lack temporal information (i.e. if the generator is not implicitly encoding the temporal information) it could result in 1) a misclassification of videos in terms of temporal location and consequently 2) a failure of the Gradient Reversal Layer (GRL) to fulfill its intended function.

---

> > ### Comment · Reviewer_kbs5 · 2023-11-23
> >
> > Thanks for the detailed response. However, many of my concerns still remain unaddressed, especially regarding the temporal classifier. Besides, while the method shows a slight enhancement in the FVD score, it does not improve the visual quality of the generated video. I prefer to keep my score.

---

> ### Author Response · Authors · 2023-11-23
> **Additional Response to Reviewer kbs5**
>
> Thank you for your comprehensive feedback. Each point raised in the initial comments has been carefully addressed, with a focus on understanding the underlying concerns. However, it appears that the concerns persist, particularly with regard to the temporal classifier. Further clarification on the specific aspects perceived as unaddressed would be greatly appreciated, enabling a more targeted and effective response.
>
> Furthermore, I acknowledge your observation about the improvement in the FVD score without a corresponding enhancement in the visual quality of the generated video. Alongside the explanation in the original manuscript, I would like to underscore that the paper addresses a specific issue in the video generation task, namely the encoding of temporal information. Consequently, a simple neural network can temporally classify the generated videos, a phenomenon we acknowledge as unconventional. To mitigate this temporal encoding, our method introduces only the Gradient Reversal Layer (GRL) with the temporal classifier during video generator training and nothing more. The outcome is an improved FVD score while maintaining visual quality equal to that of the original works. It is essential to assert that our method does not contain any components designed to enhance video generation in terms of video quality and/or FVD score. Instead, our argument centers around the alignment of generated videos with real-world characteristics, justified by the observed FVD score improvement.
>
> Adding to the reviewer's concern about the visual results, it's pertinent to note that developments in image and video generation exhibit similar trends. In image generation, the focus has shifted from creating realistic images to exploring applications like style transfer, as the difference in visual quality became less meaningful. (The visual quality was mostly indistinguishable.) Similarly, in video generation tasks, the evolution moved from generating realistic videos to realistic temporal information [2, 3, 4]. As these tasks began leveraging well-trained image generation models [1, 3], the visual differences became less significant. In line with this, our emphasis on addressing the problem in current temporal encoding in video generation is reflected only through the FVD score not by the visual enhancement, however, we believe that this research will pave the way for future video generation tasks as it, for the first time, revealed the problem in video generation task.
>
> [1] Tian, Yu, et al. "A Good Image Generator Is What You Need for High-Resolution Video Synthesis." _International Conference on Learning Representations_. 2020.
>
> [2] Yu, Sihyun, et al. "Generating Videos with Dynamics-aware Implicit Generative Adversarial Networks." _International Conference on Learning Representations_. 2021.
>
> [3] Skorokhodov, Ivan, Sergey Tulyakov, and Mohamed Elhoseiny. "Stylegan-v: A continuous video generator with the price, image quality and perks of stylegan2." _Proceedings of the IEEE/CVF Conference on Computer Vision and Pattern Recognition_. 2022.
>
> [4] Wang, Yuhan, Liming Jiang, and Chen Change Loy. "StyleInV: A Temporal Style Modulated Inversion Network for Unconditional Video Generation." _Proceedings of the IEEE/CVF International Conference on Computer Vision_. 2023.

---

### Official Review · Reviewer_LjvM · 2023-11-02

**Soundness:** 2 fair
**Presentation:** 3 good
**Contribution:** 2 fair
**Rating:** 5
**Confidence:** 5

**Summary:**

This paper demonstrates current unconditional video generation models do not considering the subtle characteristics of real-world video and proposes a simple method using Gradient Reversal Layer (GRL) with lightweight CNN to disregard the implicitly encoded temporal information within each frame. The experiment results show that neglecting implicitly encoded temporal information does not affect generated video quality and can achieve better or comparable FVD score.

**Strengths:**

This paper presents a very interesting perspective to estimate the realness of the generated video samples: the temporal locations of frames within random videos. This paper finds CNNs fail to classify the temporal locations from real-world video samples. But CNNs can precisely classify the temporal location of generated video samples. Based on this phenomenon, this paper proposes to use a lightweight CNN to disregard the implicitly encoded temporal information within each frame.

**Weaknesses:**

I agree that the videos generated by the model should strive to be as similar as possible to real-world videos in various aspects. However, I have some doubts about your design using CNNs to classify the absolute positions of each frame in a 16-frame video. The positions of video frames should be relative rather than absolute. For example, after sampling many short videos of 16 frames each from a long video, the first frame of one short video may be the last frame of another video. This might make it difficult for CNNs to classify the position of every video frame in real-world datasets. However, for videos generated by video generation models, since they are trained on short videos (e.g., 16 frames) during their training phase, it is easy for the generation model to remember the relative positions between frames. This makes it easier for CNN classifiers to classify the positions of video frames. I believe that more training on real-world datasets may improve their classification performance.

This paper employs a Gradient Reversal Layer (GRL) to weaken the temporal information in each video frame. The authors use GRL in several places, such as, "We integrate a Gradient Reversal Layer (GRL) along with an ImageNet pre-trained model," "We adopt an adversarial training technique using GRL with a simple network," "We propose a method consisting of GRL with the temporal classifier." These statements may have caused a lot of confusion for readers in understanding GRL. What exactly is GRL, and how does it function within the context of this article?

In terms of experiments, the authors do not provide a video demo to demonstrate the quality of its visual generation. I think in terms of video generation, the visual quality of the generated video results is far more important than the value of FVD.

In addition, there are some typos in the article:
The proposed method can be simply added to existing video generation methods in a plug-and-play manner. The full framework of the proposed method is shown in Fig. ??. -> In page 5

it is negligible as the difference is only 5%p. -> In page 8

**Questions:**

see weakness

---

> ### Author Response · Authors · 2023-11-17
> **Author Response to Reviewer LjvM**
>
> Thank you again for having taken the time to thoroughly evaluate our submitted manuscript.
>
> In reaction to this, we went at great lengths to thoroughly address and accommodate every single one of your comments.
>
> We would very much appreciate you going through our responses to your comments, as well as the revised version. We hope that you are able to reconsider your evaluation and potentially provide additional feedback for a final version of the paper.
>
> Many thanks in advance for your time and efforts.
>
> ---
>
> (W1) Real-world video image classification setting
>
> (Response – W1) We apologize for any confusion resulting from the lack of clarity in presenting the real-world video image classification setting. As mentioned by the reviewer, we recognized the importance of temporal classification settings between the real-world videos and the generated videos to be as same as possible. To address this, we constructed the real-world video dataset for temporal classification with the following steps. 1) We randomly choose a single content category, 2) We fix the 0th frame as the first frame for all 2,048 videos, and finally 3) We randomly sample the rest 15 frames from the same video.
> We have revised the paper by adding an explanation of the real-world temporal classification setting in the preliminary section. We believe that the reviewer’s concern that the ‘first frame of one short video may be the last frame of another video’ is resolved through step #2.
> As for the training of the real-world videos with longer epochs, we saw that training accuracy converged around 120 epochs and selected 150 epochs for temporal classification training. Therefore, we firmly state that the longer training on the real-world videos does not increase the evaluation accuracy or the training accuracy.
>
>
> (W2) Multiple usage of GRL and its meaning
>
> (Response – W2) Thank you for mentioning the part that may confuse the readers but is essential for understanding our paper. GRL implies the Gradient Reversal Layer in all of its uses in our paper. The main function of GRL is to reverse the gradient during the backpropagation process but does not change the features in the forward process. During training, the GRL backpropagates -1{\times}gradient to the generation model while backpropagating the original gradient to the temporal classifier. This makes the generation model to explicitly not capture the temporal information while the temporal classifier tries to identify the temporal features and classify them to the correct temporal location.
>
>
> (W3) Visual results
>
> (Response – W3) We indeed agree with the reviewer in viewing the visual quality over FVD in the generation task. However, as mentioned by the reviewer, the core message of the paper is the problem of implicit temporal encoding in generated videos. Additionally, we emphasize that our proposed method does not have any module or loss term to directly benefit the visual quality of the generated videos, rather, we have neglected the implicit temporal encoding when generating videos. To further demonstrate that the proposed method does not harm the visual quality of the original works while achieving higher FVD just by neglecting the temporal information encoding, we have added a few samples of generated videos in the revised appendix.
>
>
> (Additional Comments)
>
> (Response – Additional Comments) Thank you for catching the typos, we have revised the paper accordingly.

---

### Official Review · Reviewer_xNmb · 2023-11-03

**Soundness:** 2 fair
**Presentation:** 3 good
**Contribution:** 2 fair
**Rating:** 3
**Confidence:** 2

**Summary:**

This paper tackles video generation problem. The authors find that temporal information are "secretly" encoded in videos generated by existing GAN-based video generation methods. To ensure temporal information is not encoded in generated videos, the authors propose to add Gradient Reversal Layer (GRL) to video generation models. Experiments are conducted on three datasets. The proposed method outperforms MoCoGAN and achieves comparable performance with StyleGAN.

**Strengths:**

1. Although the presentation of this paper can be improved, it is quite easy to follow the main idea.

2. Comparison with 2 baselines, MoCoGan and StyleGAN are conducted on three different datasets. The proposed method outperforms MoCoGAN and achieves comparable performance with StyleGAN.

**Weaknesses:**

1. Lack of comparison with recent methods, e.g., [1, 2, 3]. These mehtods seem to have much lower FVD than the proposed methods on UCF101 dataset.

2. Marginal performance improvement. The proposed method achieves slightly better performance than its baseline, i.e., MoCoGAN, on various datasets. However, the margin is quite small, e.g., FVD of 2539 (MoCoGAN) v.s. FVD of 2360 (proposed method). Such FVD improvement may not be sufficient to convince readers that visual quality of videos generated by the proposed method is better than that of MoCoGAN.

3. I am not able to see why encoding temporal information in generated videos is a major problem that prevents GAN-based methods to generate high quality videos.

4. Minor issues (1) There are "??" in this paper. (2) The authors claim that they investigate "meaning of ‘realness’ in the video generation models" in the abstract. However, I am having a hard time finding how the meaning of 'realness' connects to the proposed method.

[1] Align your Latents: High-Resolution Video Synthesis with Latent Diffusion Models

[2] Make-A-Video: Text-to-Video Generation without Text-Video Data

[3] CogVideo: Large-scale Pretraining for Text-to-Video Generation via Transformers

**Questions:**

1. Why encoding temporal information in generated videos is a major problem that prevents GAN-based methods to generate high quality videos?

---

> ### Author Response · Authors · 2023-11-17
> **Author Response to Reviewer xNmb (1 / 2)**
>
> Thank you again for having taken the time to thoroughly evaluate our submitted manuscript.
>
> In reaction to this, we went at great lengths to thoroughly address and accommodate every single one of your comments.
>
> We would very much appreciate you going through our responses to your comments, as well as the revised version. We hope that you are able to reconsider your evaluation and potentially provide additional feedback for a final version of the paper.
>
> Many thanks in advance for your time and efforts.
>
> ---
>
> (W1) Lack of comparison with recent methods, e.g., [1, 2, 3]. These methods seem to have much lower FVD than the proposed methods on UCF101 dataset.
>
> (Response – W1) For comparison, we have selected a recent GAN-based model in the unconditional video generation field that incorporates separate content and motion noise vectors. Also, we highlight that we are not claiming to have achieved state-of-the-art performance compared to recent works. Rather, we emphasize that by neglecting implicitly encoded temporal information, we obtain a better FVD score while maintaining similar qualitative video generation results. However, we admit that experimenting with diffusion-based unconditional video generation models enriches our paper by indicating that implicit temporal encoding occurs not only with the GAN-based models but also with diffusion-based models.
> Due to the time constraint of the discussion period, we were unable to train the diffusion-based model and diffusion-based model + ours. Consequently, we focused on testing whether the generated videos from the trained diffusion-based model are temporally classifiable or not. We searched for a diffusion-based model that has open-source implementation and the checkpoint provided by the author. We were not able to find the right fit in the reviewer’s suggested methods therefore we present the experiment done on another reviewer’s suggested method, PVDM[1]. In line with our original experiments, we have fixed the content noise and varied the motion noise in PVDM. The generated videos of Sky-Timelapse and UCF-101 with PVDM do have higher temporal classification accuracy when compared to the real videos in Sky-Timelapse and UCF-101 as shown in the below table.
>
> | Dataset | Sky-Timelapse | UCF-101 |
> | --- | --- | --- |
> | Real-World Videos | 9.47% | 15.72% |
> | Generated Videos (PVDM) | 45.55% | 19.66% |
>
> Although the difference in test accuracy of UCF-101 may seem negligible, we argue that an improvement in the quality of the generated UCF-101 videos could correspond to an increase in temporal accuracy. We added qualitative and quantitative results in the appendix of the revised version as this could give insights for research in diffusion-based generation models.
>
>
> (W2) Marginal performance improvement. The proposed method achieves slightly better performance than its baseline, i.e., MoCoGAN, on various datasets. However, the margin is quite small, e.g., FVD of 2539 (MoCoGAN) v.s. FVD of 2360 (proposed method). Such FVD improvement may not be sufficient to convince readers that visual quality of videos generated by the proposed method is better than that of MoCoGAN.
>
> (Response – W2) The FVD score may not appear as appealing when considering a method designed to generate higher-quality videos, incorporating novel loss terms or modules with the goal of enhancing the video quality. However, our emphasis in this paper is on introducing a novel problem in current unconditional video generation models. Experimentally we demonstrated that neglecting this problem contributes to an improvement in the performance of video generation in terms of FVD score.
>
>
> (W3 & Q1) I am not able to see why encoding temporal information in generated videos is a major problem that prevents GAN-based methods to generate high quality videos. / Why encoding temporal information in generated videos is a major problem that prevents GAN-based methods to generate high quality videos?
>
> (Response – W3 & Q1) As implied in the manuscript, our main statement is that videos generated with current unconditional video generation methods have temporal information encoded as the reviewer said and our method ensures temporal information to be not encoded in generated videos. More importantly, we are not proposing additional loss terms or modules aiming at generating better quality videos. We demonstrated that by neglecting such temporal information encoding when training the generation model, we achieved better FVD scores while showing similar visual quality with the original works as shown in Figure 3 (FaceForensics and Sky-Timelapse) and in the revised appendix (UCF-101).
>
>
> [1] Yu, Sihyun, et al. "Video probabilistic diffusion models in projected latent space." Proceedings of the IEEE/CVF Conference on Computer Vision and Pattern Recognition. 2023.

---

> ### Author Response · Authors · 2023-11-17
> **Author Response to Reviewer xNmb (2 / 2)**
>
> (W4) Minor issues (1) There are "??" in this paper. (2) The authors claim that they investigate "meaning of ‘realness’ in the video generation models" in the abstract. However, I am having a hard time finding how the meaning of 'realness' connects to the proposed method.
>
> (Response – W4) Thank you for catching the minor mistake, the correct reference can be seen in the revised version.
> While acknowledging the challenge of precisely defining realness, we attempted to characterize it as video possessing similar attributes to those in real-world videos and with how humans perceive the temporal location within videos. To quantify this aspect, the commonly used FVD metric could be a suitable choice. However, we propose considering a temporal classification metric as well. This metric aligns with human behavior, connecting the way humans perceive videos and classify temporal location with the notion of realness.

---

> ### Comment · Reviewer_xNmb · 2023-11-23
>
> Thank you for the detailed reply and the new results.
>
> While this paper identifies an interesting potential problem of existing GAN-based video generation models for the first time, some of questions are not well addressed, e.g., W2 and W3. It seems to me that the goal is to show that the temporal information harms FVD but not visual quality. While there are evidence that supports this argument, the small FVD improvements is not convincing enough. More importantly, the goal of video generation is not to generate videos that have low FVD but to generate videos that have great visual quality. I am not sure how the findings of this paper help researchers achieve this goal.

---

> ### Author Response · Authors · 2023-11-23
> **Additional response to Reviewer xNmb**
>
> We appreciate the thorough feedback and thoughtful consideration of our latest results. Our primary goal of this paper is to demonstrate that removing the implicit encoding of temporal information leads to a decrease in FVD scores. While the evidence supports this argument, we understand your concern about the modest improvement in FVD not being convincing enough. We recognize that the standard expectation is for FVD score and visual quality to improve concurrently in video generation tasks. In our case, FVD dropped while visual quality remained consistent. We interpret this as the removal of temporal information enhancing the realism of the generated videos in the feature space without impacting the image space.
>
> Adding to the reviewer's concern about the visual results, it's pertinent to note that developments in image and video generation exhibit similar trends. In image generation, the focus has shifted from creating realistic images to exploring applications like style transfer, as the difference in visual quality became less meaningful. (The visual quality was mostly indistinguishable.) Similarly, in video generation tasks, the evolution moved from generating realistic videos to realistic temporal information [2, 3, 4]. As these tasks began leveraging well-trained image generation models [1, 3], the visual differences became less significant. In line with this, our emphasis on addressing the problem in current temporal encoding in video generation is reflected only through the FVD score not by the visual enhancement, however, we believe that this research will pave the way for future video generation tasks as it, for the first time, revealed the problem in video generation task.
>
> [1] Tian, Yu, et al. "A Good Image Generator Is What You Need for High-Resolution Video Synthesis." _International Conference on Learning Representations_. 2020.
>
> [2] Yu, Sihyun, et al. "Generating Videos with Dynamics-aware Implicit Generative Adversarial Networks." _International Conference on Learning Representations_. 2021.
>
> [3] Skorokhodov, Ivan, Sergey Tulyakov, and Mohamed Elhoseiny. "Stylegan-v: A continuous video generator with the price, image quality and perks of stylegan2." _Proceedings of the IEEE/CVF Conference on Computer Vision and Pattern Recognition_. 2022.
>
> [4] Wang, Yuhan, Liming Jiang, and Chen Change Loy. "StyleInV: A Temporal Style Modulated Inversion Network for Unconditional Video Generation." _Proceedings of the IEEE/CVF International Conference on Computer Vision_. 2023.

---

### Official Review · Reviewer_v3eo · 2023-11-08

**Soundness:** 3 good
**Presentation:** 2 fair
**Contribution:** 3 good
**Rating:** 6
**Confidence:** 3

**Summary:**

The paper discusses a phenomenon in unconditional video generation models where each frame seems to inadvertently encode information about its temporal location, which should not be the case since a single frame typically provides limited temporal context. This unintended encoding allows Convolutional Neural Networks (CNNs), which are designed to mimic aspects of human visual processing, to classify the temporal location of a video's frames accurately. To address this issue, the authors propose a new method that involves incorporating a Gradient Reversal Layer (GRL) with a lightweight CNN into existing models. The GRL layer aims to explicitly disregard the temporal information that has been implicitly encoded into frames. The authors' method was tested across various video generation models and datasets and was found to be effective in a plug-and-play fashion. The results indicated that their approach could reduce the undesired temporal information encoding without negatively affecting the Frame Video Distance (FVD) score, a common metric for video generation quality. The research suggests that temporal classification accuracy should be an additional metric to assess the performance of video generation models.

**Strengths:**

- The paper presents an Interesting phenomenon that widely used convolutional neural networks embed temporal information inside the single framework.
- The paper provides an effective method to tackle the problem it proposes and demonstrates that alleviating this artifact would lead to improved video synthesis quality.
- As deep fake is widely concerned today, this artifact this paper proposed can be served as a method of detecting the generated videos.

**Weaknesses:**

- It is not clear why encoding the temporal information inside the frame would lead to the video quality degradation. Some empirical / theoretical explanation could be useful to provide further insights.
- The argument that since CNN is inspired from humans, then they should not be able to detect the temporal signal embedded in the generated frames is not necessary. The main point of the paper tries to show that the generated videos have clear temporal information inside a single frame. The author could raise less confusion if framing the CNN detector is a simple quantitative tool to detect the time information embedded inside the video frame.
- The paper would become more appealing if framing it as a method against fake generated video and find applications in face swapping detection. Adding some experiments in this domain could make the impact broader,

**Questions:**

The authors mentioned about the human study but there is no explicit section to document the details of how to conduct the human study.

---

> ### Author Response · Authors · 2023-11-17
> **Author Response to Reviewer v3eo**
>
> Thank you again for having taken the time to thoroughly evaluate our submitted manuscript.
>
> In reaction to this, we went at great lengths to thoroughly address and accommodate every single one of your comments.
>
> We would very much appreciate you going through our responses to your comments, as well as the revised version. We hope that you are able to reconsider your evaluation and potentially provide additional feedback for a final version of the paper.
>
> Many thanks in advance for your time and efforts.
>
> ---
>
> (W1) It is not clear why encoding the temporal information inside the frame would lead to the video quality degradation. Some empirical / theoretical explanation could be useful to provide further insights.
>
> (Response - W1) The temporal information inside the frame does not lead to visual quality degradation of the video. To put it in another way, the temporal information does affect the FVD but not the visible quality of the video as can be seen in Figure 3 (FaceForensics and Sky-Timelapse) and in the revised appendix (UCF-101). In terms of FVD, we hypothesize that it could degrade due to temporal information encoding as each frame from a generated video has additional information that is not essential/needed in case of the real-world videos.
>
>
> (W2) The argument that since CNN is inspired from humans, then they should not be able to detect the temporal signal embedded in the generated frames is not necessary. The main point of the paper tries to show that the generated videos have clear temporal information inside a single frame. The author could raise less confusion if framing the CNN detector is a simple quantitative tool to detect the time information embedded inside the video frame.
>
> (Response - W2) Thank you for addressing the concern that could potentially hinder readers from understanding our paper’s core message. The initial purpose of stating ‘CNN is inspired from humans’ was to illustrate why the problem of implicit temporal information in the generated videos is an incomprehensible phenomenon. However, as we agree that argument could be minimized and revised the paper accordingly.
>
>
> (W3) The paper would become more appealing if framing it as a method against fake generated video and find applications in face swapping detection. Adding some experiments in this domain could make the impact broader.
>
> (Response – W3) Thank you for pointing out an application of the temporal classification metric in another field. To assess the effectiveness of the proposed temporal classification metric in detecting deepfake videos, we utilized the Celeb-DF[1] and Celeb-DF-v2[1] datasets which are commonly used in the field of deepfake detection. Each dataset consists of real and synthesized videos where the synthesized videos are made by face swapping the real video with another face identity with an auto-encoder. For a fair comparison between real and synthesized videos, we first randomly select a video from real videos. Then, we select the same video with another face identity in the synthesized video. In line with our original experiments, we fix the first frame with the 0th frame from each video and randomly sample the rest 15 frames from the video. We note that the 15 frames are of the same index, meaning that they are from the same temporal location.
> The temporal classification accuracy of the synthesized video is around 1.07%p higher than real video results. As the difference is not as notable as the results in the main table (with generated and real videos), the temporal classification metric cannot be directly used for detecting deepfake videos. With curiosity we have also experimented with face-cropped versions of each video, considering that synthesized videos are generated solely by face-swapping the original video. We obtained face-cropped versions utilizing the Haar Cascade algorithm. The result followed a similar trend with the initial experiment showing around 0.92%p difference. We hypothesize that the small temporal accuracy difference comes from not generating entire videos but rather only detecting and swapping faces, thereby leaving the temporal characteristics of the real video unchanged. However, as the extended experiment on the deepfake videos may give a pathway to future research in deepfake detection, we added a section in the appendix in the revised version.
>
>
> (Q1) The authors mentioned about the human study but there is no explicit section to document the details of how to conduct the human study.
>
> (Response – Q1) The user study mentioned in the manuscript is to emphasize the usage of the user study in the prior works. We have not conducted any human studies for our experiment and thus do not have a section in the original paper.
>
>
> [1] Li, Yuezun, et al. "Celeb-df: A large-scale challenging dataset for deepfake forensics." Proceedings of the IEEE/CVF conference on computer vision and pattern recognition. 2020.

---

### Author Response · Authors · 2023-11-17
**General Response to Reviewers**

Dear Reviewers, we appreciate your thorough reviews and the insightful feedback provided regarding our paper.

As many reviewers mentioned regarding the confusion in the introduction, we have revised it to focus more on the implicit temporal encoding in current unconditional video generation models and minimize on the connection between humans and CNNs. Through the experiment with the diffusion-based unconditional video generation model, we have found the diffusion-based model has similar traits of encoding temporal information in the generated videos.

Here is the list of revisions incorporated into our paper:

Modification in abstract, introduction, and preliminary.
Added a line for clarifying how the temporal classifier is trained in the method.
Inclusion of additional images (UCF-101 and sample for temporal classification), verify the existence of the problem in a diffusion-based model (PVDM[1]), and ablation study in the deepfake detection task.
Minor modifications. (i.e., typos, missing references, figure captions, etc.)

[1] Yu, Sihyun, et al. "Video probabilistic diffusion models in projected latent space." Proceedings of the IEEE/CVF Conference on Computer Vision and Pattern Recognition. 2023.

---

### Meta-Review · Area_Chair_8S7y · 2023-12-06

**Metareview:**

(a) Scientific claims and findings

The authors demonstrate that the current video generation model leaks temporal information in the generated frames. Specifically, they illustrate that classification models can accurately predict the frame index of the generated video but cannot do the same for real video. Furthermore, the authors propose applying the Gradient Reversal Layer (commonly used in domain adaptation) to mitigate the problem and show that it improves the FVD.

(b) Strength

Reviewers agree that this paper investigates an interesting problem within existing video generation models. Additionally, Reviewer "v3eo" points out a potential application in detecting generated videos.

(c) Weakness

i. The significance of this finding and improvement isn't evident, raising two concerns. First, if the hidden temporal information doesn't impact visual quality but only influences FVD, why is it considered a significant problem? Second, the minor improvement in FVD might lack practical importance.

ii. The paper lacks a robust explanation for the finding. While it presents an interesting observation, the underlying cause remains unclear. As highlighted by some reviewers, this observation might be artificial, potentially stemming from the training process, data used in the video generation models, or the temporal classifier employed.

iii. The experiments and evaluations presented are unsatisfactory. There's a notable absence of exploration into more recent video generation models, and the re-trained baseline's performance is considerably inferior to what was initially reported in the original paper.

iv. The presentation could benefit from improvement. Reviewers find the comparison to human vision confusing and inaccurate. Additionally, there are minor issues in the writing and presentation that need addressing.

v. The paper falls short in providing visual results, especially in terms of video demonstrations.

**Justification For Why Not Higher Score:**

In alignment with the reviewers' perspectives, the AC finds the practical significance of the temporal information leakage unclear. Despite its impact on the FVD score, the broader community widely acknowledges the limitations of FVD. Moreover, the absence of a robust explanation for this finding restricts its contribution, as this problem might not be universally applicable.

**Justification For Why Not Lower Score:**

N/A

---

### Decision · Program_Chairs · 2024-01-16

Reject